# Long-Term Stability of Different Kinds of Gas Nanobubbles in Deionized and Salt Water

**DOI:** 10.3390/ma14071808

**Published:** 2021-04-06

**Authors:** Yali Zhou, Zhenyao Han, Chunlin He, Qin Feng, Kaituo Wang, Youbin Wang, Nengneng Luo, Gjergj Dodbiba, Yuezhou Wei, Akira Otsuki, Toyohisa Fujita

**Affiliations:** 1College of Resources, Environment and Materials, Guangxi University, Nanning 530004, China; 1815394050@st.gxu.edu.cn (Y.Z.); 1622303001@st.gxu.edu.cn (Z.H.); helink@gxu.edu.cn (C.H.); fengqin307@gxu.edu.cn (Q.F.); wangkaituo@gxu.edu.cn (K.W.); wangyoubin@gxu.edu.cn (Y.W.); nnluo@gxu.edu.cn (N.L.); yzwei@gxu.edu.cn (Y.W.); 2Graduate School of Engineering, The University of Tokyo, Bunkyo 113-8656, Japan; dodbiba@g.ecc.u-tokyo.ac.jp; 3Ecole Nationale Supérieure de Géologie, Geo Ressources UMR 7359 CNRS, University of Lorraine, 2 Rue du Doyen Marcel Roubault, BP 10162, 54505 Vandoeuvre-lès-Nancy, France; akira.otsuki@univ-lorraine.fr; 4Waste Science & Technology, Luleå University of Technology, SE 971 87 Luleå, Sweden

**Keywords:** extended DLVO theory, mean size, zeta potential, Ostwald ripening effect, Stokes equation

## Abstract

Nanobubbles have many potential applications depending on their types. The long-term stability of different gas nanobubbles is necessary to be studied considering their applications. In the present study, five kinds of nanobubbles (N_2_, O_2_, Ar + 8%H_2_, air and CO_2_) in deionized water and a salt aqueous solution were prepared by the hydrodynamic cavitation method. The mean size and zeta potential of the nanobubbles were measured by a light scattering system, while the pH and Eh of the nanobubble suspensions were measured as a function of time. The nanobubble stability was predicted and discussed by the total potential energies between two bubbles by the extended Derjaguin–Landau–Verwey–Overbeek (DLVO) theory. The nanobubbles, except CO_2_, in deionized water showed a long-term stability for 60 days, while they were not stable in the 1 mM (milli mol/L) salt aqueous solution. During the 60 days, the bubble size gradually increased and decreased in deionized water. This size change was discussed by the Ostwald ripening effect coupled with the bubble interaction evaluated by the extended DLVO theory. On the other hand, CO_2_ nanobubbles in deionized water were not stable and disappeared after 5 days, while the CO_2_ nanobubbles in 1 mM of NaCl and CaCl_2_ aqueous solution became stable for 2 weeks. The floating and disappearing phenomena of nanobubbles were estimated and discussed by calculating the relationship between the terminal velocity of the floating bubble and bubble size.

## 1. Introduction

Nanobubbles have some unique properties, unlike conventional milli- to micro-bubbles, such as high mass transfer [1], long-term stability [2,3,4,5,6,7,8,9,10], high zeta potential, high surface to volume ratio, and generating free radicals when collapsing [11,12]. Nanobubbles can be divided into surface nanobubbles absorbed on solid surfaces and bulk nanobubbles dispersed in aqueous solutions, experiencing Brownian motion. Bulk nanobubbles have diameters of less than 1 micrometer [13]. Because of their unique physico-chemical properties, nanobubbles can be used in various application fields, e.g., improvement of plant growth and productivity [14], membranes cleaning [15,16,17], waste-water treatment [1,18,19,20,21,22], visualization improvement as the ultrasound contrast agent [23], froth flotation [24,25], improvements of methane production in the anaerobic digestion [26,27], applications in food processing [28,29] and reactions with concrete using CO_2_ nanobubbles [30,31].

Different kinds of nanobubbles have different application potentials. H_2_ nanobubble gasoline blends can improve combustion performance, compared with conventional gasoline [32]. A N_2_ nanobubble water addition can enhance the hydrolysis of waste activated sludge and improve methane production in the process of anaerobic digestion [26]. O_2_ nanobubbles produce the methane in anaerobic digestion of cellulose [33] and CO_2_ bulk nanobubbles can be used in food processing [29].

Our group also reported free radical degradation in water using different kinds of nanobubbles, i.e., H_2_ in Ar, O_2_, N_2_, CO_2_ and a mixture of H_2_ in Ar and CO_2_. The hydroxyl radical was scavenged, and the superoxide anion was diminished by mixing the carbon dioxide nanobubbles after hydrogen nanobubbles existence in the water [11]. The antioxidant effect of H_2_ nanobubbles in water was found to suppress tumor cell growth [12]. In those applications, it is important to investigate the stability and quality of nanobubbles in water, depending on time for further utilization.

Since there are so many potential applications, it is important to study fundamental characteristics and properties of bulk nanobubbles in complex solutions. The low concentration of 1 mM (milli mol/L) of NaCl could stabilize O_2_ micro- and nano-bubbles for at least one week [7] because there was a shielding effect in low concentrations of NaCl. On the other hand, the higher concentration of NaCl decreased the nanobubble concentration more quickly [7]. Nirmalkar et al. (2018) performed the calculation of the interaction energies between air nanobubbles in 0–20 mM NaCl solutions by using the Derjaguin–Landau–Verwey–Overbeek (DLVO) theory. According to their calculation, the interaction potential energy was positive above pH 4 in deionized water and it decreased with an increasing salt concentration. Beyond a certain critical concentration of NaCl (between 10 to 20 mM), the system becomes unstable [8]. Meegoda et al. (2019) calculated the electrical double layer potential between O_2_ nanobubbles in 0–0.1 M NaCl solutions and reported the stability of nanobubbles in 1 mM NaCl concentration [9]. Hewage et al. (2021) also reported the stability of air nanobubbles in 1 mM of different ion valence electrolyte solutions (i.e., NaCl, CaCl_2_, FeCl_3_) was confirmed over one week [10]. However, these research studies are only partial. In order to have a better and fuller understanding of nanobubble characteristics and have a reasonable interpretation of nanobubble behaviors in different solution environments, it is important to have a comprehensive study.

In the current study, five different kinds of nanobubbles of N_2_, O_2_, 8% H_2_ in Ar, CO_2_ and air were compared with their existence period, average size, and zeta potential, as well as their suspension pH and Eh as a function of time to be considered in several applications. The property changes of nanobubbles in the presence of salt as a function of time were observed and discussed. The stability of nanobubbles via our experimental studies was interpreted and discussed by using the extended DLVO theory.

## 2. Materials and Methods

### 2.1. Materials

Deionized water with a resistivity of 18.2 MΩcm prepared by the Classic Water Purification System from Hitech instruments CO., Ltd. (Shanghai, China) was used for all the experiments. Different gases (N_2_, O_2_, Ar + 8%H_2_ (8% H_2_ and 92% Ar mixed gas), air and CO_2_) were used to prepare nanobubble water. Sodium chloride (NaCl), calcium chloride (CaCl_2_), aluminum chloride hexahydrate (AlCl_3_.6H_2_O) were used to prepare 1 mM salt solutions and nanobubble water in order to study the nanobubbles’ stability in the case of salt addition, considering the Schultze Hardy rule, i.e., $CCC\propto \;\frac{1}{z^{6}}$ where *CCC* is the critical coagulation concentration, and *z* is the ionic valence [34]. Dilute HCl or NaOH was used to adjust the pH of the nanobubble water. A 10 L glass container was used to store the nanobubble water temporarily and 50 mL plastic centrifuge tubes were used to store the nanobubble water after they were rinsed well with deionized water.

### 2.2. Generation of Different Kinds of Nanobubbles

Nanobubble water was produced by a ultra-micro bubble generator (XZCP-K-1.1) (100 nm to 10 μm) provided by Xiazhichun Co. Ltd. (Kunming, China). When the generator was turned on, gas from a gas cylinder was pumped into the machine in a negative pressure, and inlet and outlet pipes were immersed in deionized water/electrolyte solutions in the 10 L glass container where there was a circulation between the inlet and outlet, as shown in Figure 1A. The machine mixed the gas and liquid, and then high-density, uniform and “milky” nanobubble water (Figure 1B) was produced through a nozzle by the hydrodynamic cavitation method. The cloudy and milky nanobubble water gradually became clear through the process of the microbubbles rising and collapsing at the air–water interface (Figure 1C). This process took 2 to 3 min (Figure 1D). The machine worked for 15 min to generate high number density nanobubble water of more than 10^8^ bubbles/mL (obtained from NanoSight, NS300, Malvern (Worcestershire, UK) outsourcing).

After the above-mentioned preparation steps, the nanobubble water had a temperature of about 313 K. It was then left to cool down to room temperature. Next, from the prepared nanobubble water, the large sizes of the bubbles, from 1 to 10 μm, were removed by centrifugal treatment in the following manner. The nanobubble water was stored in a 50 mL centrifuge tube; then, centrifugal treatment was performed in 6000 rpm, i.e., 31.4 m/s peripheral velocity, for 6 min to remove possible impurities and large bubbles. After centrifugation, the size distribution, zeta potential, Eh and pH were measured from one centrifuge tube to record the first day’s data while other sealed centrifuge tubes with nanobubble water were kept at room temperature (298 K) for continuous measurements.

### 2.3. Characterization of Bulk Nanobubble Suspensions

The nanobubble size was measured by the dynamic light scattering method (DLS, NanoBrook Omni, Brookhaven Instruments, Holtsville, NY, USA). The machine’s measurement range is between 1 nm and 10 μm. Since nanobubbles experience Brownian motion, the scattered light fluctuates as a function of time due to the particles’ random movements, and the diffusion coefficient can be obtained from a computer digital correlator. The particle hydrodynamic diameter can be calculated by the Stokes–Einstein equation [35] as described below:(1)$$D_{T}=\frac{KT}{3\pi \eta d_{h}}$$
where *D**_T_*, *k*, *T*, *η*, *d**_h_* is the diffusion coefficient, Boltzmann constant, liquid absolute temperature, viscosity, and hydrodynamic diameter, respectively.

One measurement duration time was 3 min, and the instrument performed the size measurement 3 times. After size measurements, the zeta potential was measured by the same instrument that performed the 3 measurements, consisting of 20 runs/measurement. Every experiment was performed in duplicate and then the average size and measurement error were calculated and plotted.

The zeta potential value was obtained through the micro-electrophoresis method by the phase analysis light scattering method (NanoBrook Omni, Brookhaven Instruments). The zeta potential of the bubbles in the aqueous solution with salt was calculated by using the Smoluckowski model *f*(*κa*) = 1 in Equation (2) for *a* >> 1/*κ* (*κa* >> 1); however, the zeta potential of the bubbles in deionized water was calculated by using the Hückel model *f*(*κa*) = 2/3 in Equation (2) individually for *a* << 1/*κ* (*κa* << 1) [34] because of the small bubble size. The electrophoretic mobility *μ* in the Henry equation is shown as follows:(2)$$\mu =\frac{\varepsilon_{r}\varepsilon_{0}\zeta }{\eta }f\left(\kappa a\right)$$
where *ε_r_*, *ε*_0_ are relative permittivity and permittivity of free space, respectively, *ζ* is the zeta potential, *η* is fluid viscosity, *κ* is the Debye length, *a* is bubble radius and *f*(*κa*) is the Henry function [36].

The viscosity, refractive index and relative dielectric constant of water at 293 K are 1.002 × 10^−2^ poise, 1.332, and 80.2, respectively [37]. Nanobubble water pH and Eh values were measured by pH meter (Inlab Expert Pro, Mettler Toledo, Greifensee, Switzerland) and Eh meter (by oxidation-reduction potential meter, Inlab Redox, Mettler Toledo), respectively.

## 3. Results and Discussion

### 3.1. Zeta Potential of Different Kinds of Nanobubbles

The zeta potentials of prepared different kinds of nanobubbles (N_2_, O_2_, Ar + 8%H_2_, air and CO_2_) depending on pH were measured and are plotted in Figure 2. The pH was adjusted from the natural pH by adding NaOH or HCl to achieve the desired pH. The natural pH of nanobubbles in deionized water was between a pH of 6 to 7, except for CO_2_ nanobubble suspensions. The zeta potential of N_2_, O_2_, Ar + 8%H_2_ nanobubbles were negative at about −15 to −30 mV at a natural pH of 6 to 7, while the zeta potential of CO_2_ nanobubbles in deionized water was positive at about +10 mV at a natural pH of 4 to 4.5. The solubility of gas, dielectric constant of gas [37] and the isoelectric point (IEP) of nanobubbles determined in this study are listed in Table 1. Since the solubility of CO_2_ is high (7.07 × 10^−4^, Table 1), the HCO_3_^−^ adsorption on the CO_2_ bubble surface can cause a positive zeta potential in the deionized water due to the concentration of counter ions. The CO_2_ solubility phenomena can be explained in the following Equations (3)–(8) [38] assuming that CO_2_ gas is dissolved in water.
(3)$$CO_{2}+H_{2}O⇌H_{2}CO_{3}$$
(4)$$\frac{H_{2}CO_{3}}{CO_{2}}=1.7\times 10^{-3}\;\mathrm{at}\;298\;K$$
(5)$$H_{2}CO_{3}⇌H^{+}+HCO_{3}^{-}$$
(6)$$K_{a_{1}}=\frac{\left[H^{+}\right]\left[HCO_{3}^{-}\right]}{\left[CO_{2}\right]+\left[H_{2}CO_{3}\right]},\;pK_{a_{1}}=6.35$$
(7)$$HCO_{3}^{-}⇌H^{+}+CO_{3}^{2-}$$
(8)$$K_{a_{2}}=\frac{\left[H^{+}\right]\left[CO_{3}^{2-}\right]}{\left[HCO_{3}^{-}\right]},\;\;pK_{a_{2}}=10.33$$

The natural pH of CO_2_ nanobubbles in the deionized water was around 4.2. The IEPs of nanobubbles in the deionized water containing air, Ar + 8%H_2_, O_2,_ N_2_, and CO_2_ had a pH of 4.2, 4.6, 4.7, 5.2 and 5.7, respectively (Table 1). They were determined from the two pH values of zeta potential, assuming a linear relationship. Among all the gases studied, the IEP of CO_2_ was the highest (i.e., 5.7). This can be explained by the dissolution of the bubble surface, as discussed above, and because the IEP increased with the dielectric constant of gas increase, as listed in Table 1.

### 3.2. Time Effect of Mean Size and Zeta Potential of Different Gas Nanobubbles and pH and Eh of Nanobubble Suspensions

The effect of time after the preparation of the nanobubble water on the nanobubble mean diameter of N_2_, O_2_, Ar + 8%H_2_, air and CO_2_ gas in deionized water (A), 1 mM NaCl aqueous solution (B), 1 mM CaCl_2_ aqueous solution (C) and 1 mM AlCl_3_ aqueous solution (D) are shown in Figure 3. The zeta potential of the nanobubbles and the pH and Eh as a function of time are shown in Figure 4 and Figure 5, respectively. In this section, we introduce and discuss the results in the absence of salt followed by the results in the presence of salt. The Eh value change can be correlated to the H_2_ nanobubble concentration change, O_2_ nanobubble existence and CO_2_ nanobubble dissolution in water.

#### 3.2.1. Nanobubble Characteristics Change as a Function of Time in Deionized Water

In the absence of salt, the initial mean diameters of the N_2_, O_2_, Ar + 8%H_2_ and air nanobubbles were around 200 nm and gradually increased to 400–530 nm in 30 days. After 30 days, the mean diameter slightly decreased and was stable at 330–480 nm in 60 days (Figure 3A). The zeta potentials of those gases were between −12 and −32 mV for 60 days (Figure 4A). Variation trends of nanobubbles are similar to previous reports. Ulatowski et al. (2019) reported the O_2_ nanobubbles size produced by porous tube type was 200 nm on the initial day and unchanged after 10 days, while the N_2_ nanobubble mean size was 300 nm at −11 mV of zeta potential on the initial day and 400 nm at about −12 mV after 35 days [3]. Meegoda et al. (2019) reported the O_2_ nanobubble mean size produced by the cavitation method was 180 nm at −20 mV of zeta potential on the initial day and the O_2_ bubble mean size increased to 285 nm at −16 mV after 7 days [9]. Michailidi et al. (2020) reported that with the nanobubbles prepared by the high shear cavitation method, initially the O_2_ nanobubble mean size was 350 nm and the air nanobubble mean size was 430 nm, while the O_2_ and air nanobubble sizes increased to 560 and 500 nm, respectively, after two weeks; both bubbles existed at 640 nm size after 2 and 3 months [4]. Although the nanobubble size depends on the preparation methods, the N_2_ and O_2_ nanobubbles’ size increase with time was similar to our results that the prepared nanobubble sizes of various gases were about 200 nm initially and the nanobubble size increased to 300 to 500 nm with 2 to 4 weeks. Nirmalkar et al. (2018) reported that the number density of air nanobubbles prepared by the acoustic cavitation method gradually decreased; however, about 90 nm of nanobubbles existed for almost one year [6].

On the other hand, the CO_2_ nanobubble size gradually increased with time and the size changed from 180 nm initially to 350 nm in 5 days (Figure 3A). After 5 days, the CO_2_ nanobubbles were not stable and could not be detected by DLS because the CO_2_ dissolved in the water and the electric double layer was compressed; therefore, the bubbles were coagulated due to the Van der Waals interaction (and the hydrophobic interaction) as discussed in the following Section 3.3.1. Ushikubo et al. (2010) reported that with the nanobubbles prepared by the pressurized type, the zeta potential of CO_2_ nanobubbles was negative at pH 4, initially; however, the bubbles disappeared at pH 6.1 due to the dissociation of CO_2_ [39]. In our experiment, the zeta potential of CO_2_ nanobubbles was +9 mV at pH 4.2 on the first day and it gradually increased with time to +17 mV at pH 4.6 after 5 days (Figure 4A). As shown in Figure 5A, the Eh of Ar + 8%H_2_ nanobubble suspension increased from 250 to 400 mV in one day and also the Eh of N_2_ nanobubble suspension increased from 330 to 420 mV in one day. The Eh of O_2_ and air nanobubble suspensions were between 400 and 450 mV initially, then to 60 days. The Eh of the CO_2_ bubble suspension increased as the pH increased with time (Figure 5A’).

#### 3.2.2. Nanobubble Characteristics Change as a Function of Time in Salt Aqueous Solutions

In the presence of salt, the nanobubbles were prepared in 1 mM NaCl, CaCl_2_ or AlCl_3_ aqueous solution. In 1 mM NaCl aqueous solution, N_2_ and air nanobubble sizes were initially 200 nm and gradually increased; they were not observed after 7 days. Meanwhile, O_2_, Ar + 8%H_2_, and CO_2_ bubble sizes were initially 200 to 300 nm, gradually increased and they were not detected after 14 days (Figure 3B). The existence period of CO_2_ bubbles became longer in the 1 mM NaCl aqueous solution than in the absence of salt (i.e., 5 days in deionized water) (Figure 3A). The zeta potentials of N_2_, O_2_, Ar + 8%H_2_ and air nanobubbles were small, within ±5 mV at initial days to 14 days (Figure 4B), while the zeta potential of CO_2_ nanobubbles was higher than +12 mV initially and was kept +10 mV for 14 days (Figure 4B). Ke et al. (2019) reported that with the nanobubbles produced by the pressurized type, the initial N_2_ nanobubble average size was about 200 nm in the 0.1 mM NaCl aqueous solution [2] and the size was similar to our result in the 1 mM NaCl condition (Figure 3B). In 1 mM NaCl aqueous solution, Leroy et al. (2012) measured that the zeta potential of H_2_ gas was −20 mV at pH 6 [40] and Yang et al. (2001) reported the initial zeta potential of H_2_ nanobubbles was −30 mV [41]. In our measurement of Ar + 8%H_2_ nanobubble, the zeta potential was much smaller at −5 mV at pH 6 in 1 mM NaCl (Figure 3B). Meegoda et al. (2019) reported that the O_2_ nanobubble mean size prepared by the high shear cavitation type was 214 nm at −22 mV of zeta potential initially and the O_2_ bubble mean size was almost the same at 219 nm at −14 mV in 7 days [9]. In our experiment, the O_2_ bubble mean size was about 300 nm for 14 days (Figure 3B), similar to the above-mentioned previous result; however, the zeta potential was +6 mV (Figure 4B). The Eh value increased from 220 to 360 mV with the Ar + 8%H_2_ nanobubble suspension in one day, which was the same as nanobubbles in the deionized water suspension. The Eh of the N_2_ nanobubble suspension was 360 mV and the Eh of other nanobubble suspensions was a little high at about 450 mV in 7 days (Figure 5B). The pH of CO_2_ was from 4.3 to 4.4 and the Eh, pH, zeta potential and mean size of the CO_2_ suspension were almost constant in the 1 mM NaCl aqueous solution compared with its suspension in deionized water.

In 1 mM CaCl_2_ salt aqueous solution, O_2_ was not stable after 7 days, while N_2_, Ar + 8%H_2_ and air bubbles could be observed until 14 days, but no more were detected after 14 days (Figure 3C). The CO_2_ nanobubbles existed until about 20 days in 1 mM CaCl_2_. The absolute value of the zeta potential of N_2_, O_2_, Ar + 8%H_2_ and air nanobubbles was low, at less than 10 mV, while the zeta potential of CO_2_ was higher than +10 mV (Figure 4C). Cho et al. (2005) reported that the air nanobubble size prepared by the sonicating method was 850 nm at −8 mV zeta potential [42], and their reported results were similar to our results after one day (Figure 3C and Figure 4C). The size of the air, N_2_ and O_2_ nanobubbles in 1 mM CaCl_2_ salt aqueous solution increased extremely with time. Yang et al. (2001) reported that the zeta potential of the H_2_ nanobubble was −5.5 mV [8], and their result was similar to our result of Ar + 8%H_2_ nanobubble after one day (Figure 4C). For the Eh value (Figure 5C), the Eh of the 8%H_2_+Ar nanobubble suspension increased from 190 to 370 mV in one day, same as the nanobubbles in the other water suspensions. The Eh of N_2_, O_2_, air and CO_2_ nanobubble suspensions were between 400 and 450 mV and the pH of N_2_, O_2,_ Ar + 8%H_2_ and air nanobubble suspensions were between 5.5 and 6. The pH of CO_2_ solution increased from 4.4, initially, to 5.3 after 7 days (Figure 5C).

In 1 mM AlCl_3_ salt aqueous solution, N_2_ and O_2_ were not stable after 7 days, while air, CO_2_ and Ar + 8%H_2_ nanobubbles could be observed until 14 days; however, they were not detected after 14 days. The size of the Ar + 8%H_2_ nanobubble was more stable, at between 200 and 300 nm for 14 days, compared with other bubbles (Figure 3D). The zeta potentials of all prepared bubbles were positive (Figure 4D) possibly due to the presence of high valence ions (Al^3+^) adsorbing on the bubble surfaces. Yang et al. (2001) reported that the zeta potential of H_2_ nanobubble was +12 mV [41], and their result was similar to our result of +15 mV of Ar + 8%H_2_ nanobubbles at the initial day (Figure 4D). Han et al. (2006) reported that the zeta potential of O_2_ nanobubbles prepared by electrolysis was +20 mV at pH 6 in 10 mM NaCl + 1 mM AlCl_3_ aqueous solution [43]. The trivalent ion Al^3+^ ion caused the positive charge in all our prepared nanobubbles (Figure 4D). For the Eh value (Figure 5D), the Eh of the Ar + 8%H_2_ nanobubble suspension increased from 190 to 410 mV in one day, same as the gas nanobubbles in the other water suspensions. The Eh of N_2_, O_2_, air and CO_2_ nanobubble suspensions were between 450 and 500 mV, and were higher than the Eh of the Ar + 8%H_2_ suspension. The pH of N_2_, O_2,_ Ar + 8%H_2_ and air were from 4.15 to 4.35, which were lower than the pH of other suspensions. The pH of CO_2_ slightly increased from 4.0 to 4.3 in 5 days (Figure 5A’) and its increase was limited compared with other suspensions in the salt aqueous solution and deionized water (Figure 5B).

The stable days of N_2_, O_2_, 8% H_2_+Ar, air and CO_2_ nanobubbles produced in different solutions are listed in Table 2. The N_2_, O_2_, Ar + 8%H_2_ and air nanobubbles in 1 mM of the three salts with different valences (+1, +2 and +3) decreased the existence period for 1 to 2 weeks compared with deionized water in the absence of salt (>60 days). On the other hand, CO_2_ bubbles existed only 5 days in deionized water in the absence of salt; however, CO_2_ bubbles existed 14 days in the 1 mM salt aqueous solution. The stability differences will be further discussed in the following section.

### 3.3. Nanobubble Stabilization Estimation by Using the Extended DLVO Theory

The thickness of the electric double layer (Debye length = 1/*κ*) of the nanobubble is calculated in the next formula:(9)$$\kappa =\sqrt{\frac{2nz^{2}e^{2}}{\varepsilon_{r}\varepsilon_{0}kT}}$$
where *κ* is the Debye–Hückel parameter, *n* is the concentration of anion or cation in the solution and is equal to 1000 *N_A_C* (*N_A_* is the Avogadro number and *C* is concentration mol/L), *z* is the valence of ion, *e* is the electron charge, *ε_r_* is the relative dielectric constant and *ε*_0_ is the permittivity of vacuum.

The calculated Debye lengths of the nanobubbles studied in our experiment are listed in Table 3 in order to discuss differences in the nanobubble stability and to have some idea of how to keep the nanobubble stable for a longer duration. The Debye length is 300 nm at pH 6 around N_2_, O_2_ and Ar + 8%H_2_ nanobubbles in deionized water, and 140 nm at pH 4.2 and 70 nm at pH 4.7 around CO_2_ nanobubbles. Since the characteristics of CO_2_ nanobubbles are different from others (Figure 3, Figure 4 and Figure 5), the Debye length (1/*κ*) around CO_2_ bubbles and HCO_3_^−^, OH^−^, CO_3_^2−^ ion concentration in the presence of CO_2_ nanobubbles in deionized water as a function of pH were also calculated and shown in Figure 6 in order to discuss their relationship. The concentration of various ions was calculated from using Equations (3)–(8).

A thick electric double layer (300 nm, Table 3) is present around the N_2_, O_2_ and Ar + 8%H_2_ nanobubbles in deionized water and can stabilize those nanobubbles with high enough zeta potential (i.e., −16 to −32 mV, Figure 4A). On the other hand, the Debye length of CO_2_ nanobubbles (70 nm at pH 4.7 in DI water, Table 3) and other nanobubbles in salt aqueous solution (3–10 nm, Table 3) are thin and thus, the influence of the Van der Waals and hydrophobic attraction can be more dominant in coagulating and coalescing the nanobubbles. In order to quantify the influence of the electrical double layer potential, the Van der Waals potential and the hydrophobic interaction potential, their potential energies were calculated by using the extended DLVO theory as described in the following sections. Tchaliovska et al. investigated the thickness of thin flat foam films formed in aqueous dodecyl ammonium chloride solution [44]. Angarska et al. showed that the value of the critical thickness of the foam film was sensitive to the magnitude of the attractive surface forces acting on the film using sodium dodecyl sulfate and the total disjoining pressure operative on the films, which was expressed as a sum of the Van der Waals and hydrophobic contributions [45].

Figure 7 shows the two nanobubble positions and geometries considered in our potential calculation. The total potential energy *V_T_* between two nanobubbles can be given by the potential energy due to the Van der Waals interaction (*V_A_*), hydrophobic interaction (*V_h_*) and the electrostatic interaction (*V_R_*) as follows [46,47,48,49,50];
(10)$$V_{T}=V_{A}+V_{h}+V_{R},\;\frac{V_{T}}{kT}=\frac{V_{A}+V_{h}+V_{R}}{kT}$$
*V_A_* + *V_h_* is shown in the follow equation:(11)$$V_{A}+V_{h}=-\frac{A+K}{6}\left[\frac{2a_{1}a_{2}}{R^{2}-{\left(a_{1}+a_{2}\right)}^{2}}+\frac{2a_{1}a_{2}}{R^{2}-{\left(a_{1}-a_{2}\right)}^{2}}+\ln\left(\frac{R^{2}-{\left(a_{1}+a_{2}\right)}^{2}}{R^{2}-{\left(a_{1}-a_{2}\right)}^{2}}\right)\right]$$
where the Hamaker constant *A* for air in water whose value of air–water–air of 3.7 × 10^−20^ J [46] was used to investigate our system in nanobubble–water–nanobubble. The *K* is a hydrophobic constant for air in water. Yoon and Aksoy calculated the *K* using the extended DLVO theory [49] while Wang and Yoon measured the *K* as a function of the SDS concentration at different kinds of NaCl concentrations and showed that *K* was estimated at 10^−17^ J in the absence of salt and 10^−19^ J in the 1 mM NaCl aqueous solution [50]. In our calculation, 10^−19^ J of *K* was used in 1 mM CaCl_2_ and AlCl_3_ aqueous solutions and 10^−18^ J of *K* was used in deionized water containing CO_2_ nanobubbles by considering the reference values [50].

The radius of nanobubbles is *a*_1_ and *a*_2_ and their distance *R* = *a*_1_ + *a*_2_ + *h* is defined as shown in Figure 7, and *h* is the surface-to-surface distance between two nanobubbles. The potential energy of *V_R_* is expressed as follows,
(12)$$V_{R}=-\frac{\pi \varepsilon_{r}\varepsilon_{0}a_{1}a_{2}\left(\psi_{1}^{2}+\psi_{2}^{2}\right)}{a_{1}+a_{2}}\left[\frac{2\psi_{1}\psi_{2}}{\psi_{1}^{2}+\psi_{2}^{2}}ln\frac{1+\exp\left(-\kappa h\right)}{1-\exp\left(-\kappa h\right)}+\ln\left\{1-\exp\left(-2\kappa h\right)\right\}\right]$$
where $\psi_{1}\;and\;\psi_{2}\;$ are surface potentials of the nanobubbles of radii *a*_1_ and *a*_2_, respectively.

#### 3.3.1. Stability Calculation of Nanobubbles in Deionized Water

The mean diameters of the prepared nanobubbles of N_2_, O_2_, Ar + 8%H_2_ and air in deionized water were between 170 and 230 nm, as shown in Figure 3. With time, these diameters gradually increased, and they were between 390 and 530 nm after 30 days. Then, they slightly decreased to between 330 and 480 nm and were stable after 60 days. As the mean bubble diameter was measured in our experiment, the increase in bubble size indicates that the small size nanobubbles coalesce with other bubbles, and those small bubbles disappear.

The above-mentioned phenomenon is well known as the Ostwald ripening effect [51], which explains the deposition of a smaller object on a larger object due to the latter being more energetically stable than the former. By using Lemlich’s theory [52], it can be explained by using next equation [53]:(13)$$\frac{da}{dt}=K^{\prime }\left(\frac{1}{p}-\frac{1}{a}\right)$$
where *a* is the bubble radius, *t* is time, *K′* is a constant and *p* is instantaneous bubble mean radius. This equation tells us that the bubbles with a radius larger than *a* grow, and those with a radius smaller than *p* shrink. By using the results shown in Figure 2 (zeta potential) and Figure 3 (bubble size), the bubble stability is further discussed by calculating the total potential energy between two nanobubbles.

Figure 8 shows the potential energies between two nanobubbles in deionized water. Figure 8A shows the potential energies between two same-size gas bubbles of 200 nm of Ar + 8%H_2_ on the initial day. The total potential energy *V_T_* is 20 kT at 300 nm (the Debye length) and the maximum potential is 30 kT. The other bubbles of air, N_2_ and O_2_ bubble show almost same potential curves. It indicates that the two bubbles have enough of a potential barrier to disperse each other. Figure 8B shows the potential energies between 450 nm of two same-size bubbles of Ar + 8%H_2_ after 30 days. The maximum total potential energy *V_T_* is 15 kT, i.e., the threshold total potential energy to determine coagulation or dispersion [54], and it indicates that Ar + 8%H_2_ nanobubbles would be stable, although the absolute value of zeta potential is smallest in Ar + 8%H_2_ nanobubbles (−13 mV, Figure 4A) compared with the ones of other O_2_, N_2_ and air nanobubbles bubbles, shown in Figure 4.

When the CO_2_ nanobubbles were prepared, their mean size was 160 nm and zeta potential was low, i.e., +9 mV. Figure 9A shows the potential energies between two 160 nm bubbles. The total potential barrier was not appeared at 140 nm the Debye length. Figure 9B shows the potential energies between two 350 nm bubbles after 5 days. As the total potential barrier was also not appeared, these larger bubbles would also coagulate, coalesce and increase the size and, thus, CO_2_ bubbles could not be observed after 5 days.

#### 3.3.2. Stability Calculation of Nanobubbles in Salt Aqueous Solutions

In the 1 mM NaCl aqueous solution, the absolute values of the zeta potential of O_2_, N_2_, Ar + 8%H_2_ and air nanobubbles were less than 10 mV; the total potential energy barrier could not appear. Figure 10A shows the potential energies between two 160 nm N_2_ nanobubbles. The total potential energy *V_T_* is negative at any distance and does not show the potential barrier; therefore, the nanobubbles are not stable and can coagulate. Figure 10B shows the potential energies between two 200 nm CO_2_ nanobubbles. Although the absolute zeta potential was slightly larger than the sN_2_ nanobubbles at the Debye length 10 nm, there is no potential barrier indicating the attraction between the two bubbles. The CO_2_ nanobubble size existed between 200 and 400 nm for 14 days, as shown in Figure 3B; however, after 14 days, the CO_2_ bubble was disappeared.

In the 1 mM CaCl_2_ aqueous solution, as the absolute values of the zeta potential of O_2_, N_2_, Ar + 8%H_2_ and air nanobubbles were less than 10 mV (Figure 4C), like the one in 1 the mM NaCl aqueous solution (Figure 4B), the total potential energy barrier did not appear. Figure 11A shows the potential energies between 230 nm of two same-size N_2_ nanobubbles on the prepared day. The total potential energy *V_T_* was negative at any distance and did not show the potential barrier, and thus it indicates that the nanobubbles were not stable due to their coagulation. The zeta potential of CO_2_ was more than +10 mV, as shown in Figure 4C; however, the potential barrier still did not appear because the Debye length around the CO_2_ bubbles decreased in the presence of 1 mM CaCl_2_ salt (i.e., 5 nm vs. 40 nm (pH 4.2) in deionized water, Table 2, while the CO_2_ nanobubble mean size 1 mM CaCl_2_ was stable at about 300 nm until 14 days (Figure 3C), same as the one in the 1 mM of NaCl aqueous solution (Figure 3B). Figure 11C shows the potential energies between two 750 nm N_2_ nanobubbles after 3 days in a 1 mM CaCl_2_ aqueous solution. With the smaller zeta potential −7 mV, the potential barrier is not appeared, and it corresponds to the further coagulation/coalescence of the bubbles (Figure 3C).

In the 1 mM AlCl_3_ aqueous solution, Figure 11B shows the potential energy between two 170 nm Ar + 8%H_2_ nanobubbles on the prepared day. Although the absolute value of the zeta potential was higher (+16 mV, Figure 4D) than the one of Ar + 8%H_2_ nanobubbles prepared in a CaCl_2_ aqueous solution (−2 mV, Figure 4C), there was no potential barrier due to the small electrostatic interaction potential (<15 kT) and the bubbles may be unstable. In Figure 11D, with the N_2_ nanobubble size of 300 nm and the higher zeta potential (+25 mV, Figure 4D), the potential barrier was not also appeared.

In the case of N_2_, O_2_, Ar + 8%H_2_ and air in 1 mM salt concentration, the presence of Ca^2+^ ion from CaCl_2_ salt decreased the absolute value of the zeta potential by its adsorption on the bubble surface and a stronger coagulation could happen by thinning the electric double layer at the natural pH (without pH adjustment). Here, the overall bubble stability phenomena are summarized. Except for the 1 mM Ca^2+^ ion aqueous solution, the stability phenomena nearly followed the Shultz–Hardy rule. Comparing bubble size and existence time, the stability order of the N_2_, O_2_, Ar + 8%H_2_ and air nanobubbles identified from our study was: no salt addition > 1 mM NaCl > 1 mM AlCl_3_ > 1 mM CaCl_2_ in deionized water. On the other hand, the stability order of CO_2_ nanobubbles was: 1 mM NaCl > 1 mM CaCl_2_ > 1 mM AlCl_3_ > no salt addition, due to the effect of CO_2_ dissolution in deionized water, as discussed in Section 3.3.

#### 3.3.3. Nanobubble Movement in Salt Aqueous Solution

The nanobubbles move by Brownian motion. The Brownian diffusion can be described in the following Equations (14) and (15) where the bubble movement distance is $\sqrt{\bar{\Delta x^{2}}}$ and Equation (1) is incorporated to define *D_T_*.
(14)$$\bar{\Delta x^{2}}=2D_{T}t=2tkT/3\pi \eta d_{h}$$
(15)$$\sqrt{\bar{\Delta x^{2}}}=\sqrt{\frac{2tkT}{3\pi \eta d_{h}}}$$

If the bubble size *d**_h_* is small, the movement distance becomes longer since they have inverse correlation, as described in Equations (14) and (15).

On the other hand, a large, coalesced bubble floats, following the terminal velocity *u* defined by the Stokes equation of laminar flow:(16)$$u=\frac{d_{h}^{2}\rho g}{18\eta }$$
where *ρ* is the density of water and *g* is the gravitational acceleration.

Bubbles experiencing diffusion change their movement direction frequently in a short time. On the other hand, the bubble displacement due to the buoyancy force is always pointing upwards against the gravitational force. The terminal velocity depends on the bubble diameter, as shown in Figure 12, based on our calculation using Equation (16). A bubble smaller than 1 μm can experience very low terminal velocity (<1 × 10^−6^ m/s) which prevents bubbles from floating against the buoyancy force. Our results partially agree with the previous literature. Nirmalkar et al. (2018) reported that the number density of about a 100 nm mean size of nanobubbles gradually decreased over one year, and still existed at about 100 nm after one year [8].

Figure 13 shows the schematic diagrams of nanobubble stability in deionized water (Figure 13A) and the nanobubble size change in a salt aqueous solution (Figure 13B). The nanobubbles in deionized water stably exist for two months by the Brownian motion. On the other hand, in the nanobubbles in a 1 mM salt aqueous solution, the nanobubbles coalesced with each other and increased the size, and the large-size bubbles gradually floated and disappeared after one or two weeks.

The surface charge on the front direction of a floating larger bubble can decrease, while the tail direction of the bubble retains more ions; thus, the electric double layer charge distribution can be distorted [55]. The bubbles in the direction of a floating large bubble can interact with that large bubble. If there is a small nanobubble in the front direction of a moving bubble, it can coagulate and coalesce with large bubbles and the coalesced large bubbles can be disappeared. This phenomenon occurred for the larger size of the bubbles in a salt aqueous solution with time, as observed in this study.

## 4. Conclusions

This research investigated five kinds of gas nanobubbles (N_2_, O_2_, Ar + 8%H_2_, air and CO_2_) for their long-term stability, to consider the application of the nanobubbles containing aqueous solutions. Each gas in a cylinder was injected into deionized water or a 1 mM of salt aqueous solution, and nanobubbles were prepared by the hydrodynamic cavitation method. The mean diameter, zeta potential of bubbles, pH and Eh of nanobubble suspensions were measured and these characteristics changed, as the function of time was also studied.
The IEPs of different gas nanobubbles in deionized water varied, and the CO_2_ nanobubbles showed the highest value of pH at 5.7.The N_2_, O_2_, Ar + 8%H_2_ and air nanobubbles in deionized water showed the long-term stability for 60 days. During the 60 days, the bubble size gradually increased and decreased. Thus, this size change, explained by the Ostwald ripening effect, was also coupled with a bubble stability discussion using the total potential energy between two nanobubbles under different conditions calculated by the extended DLVO theory.The CO_2_ nanobubbles produced in deionized water were not stable and disappeared after five days. The CO_2_ nanobubbles in water dissolved the HCO_3_^−^ ion, which could decrease the total potential energy between CO_2_ bubbles, and thus the CO_2_ nanobubbles became unstable.The N_2_, O_2_, Ar + 8%H_2_ and air nanobubbles produced in the 1 mM salt aqueous solution were not stable. The potential barrier between the nanobubbles disappeared, and the bubble size gradually increased with their coalescence, followed by floating and disappearing after 14 days for O_2_, Ar + 8%H_2_ and air nanobubbles, and 7 days for N_2_ nanobubbles. On the other hand, the CO_2_ nanobubbles in the 1 mM of NaCl and CaCl_2_ aqueous solution became more stable than the CO_2_ nanobubbles in deionized water, and a 200 to 300 nm mean bubble size was kept for two weeks.A bubble smaller than 1 μm can experience very low terminal velocity (<1 × 10^−6^ m/s) which prevents bubbles from floating against the buoyancy force.

## Figures and Tables

**Figure 1 materials-14-01808-f001:**
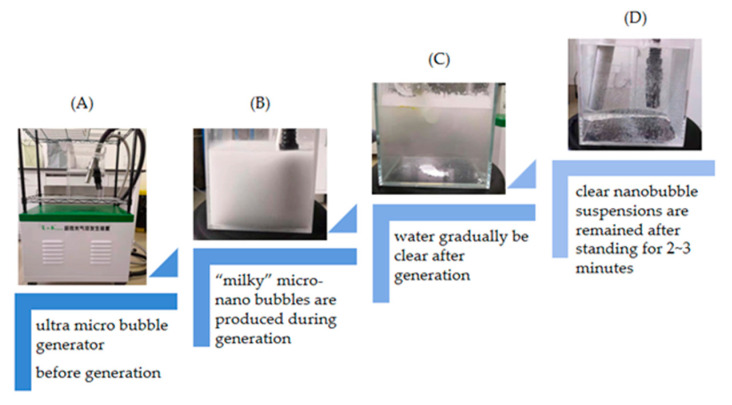
Procedure of nanobubble generation. (**A**) Before generation; (**B**) During generation of micro-nano bubbles; (**C**) Stop the generation of bubbles; (**D**) After standing for 2 to 3 min.

**Figure 2 materials-14-01808-f002:**
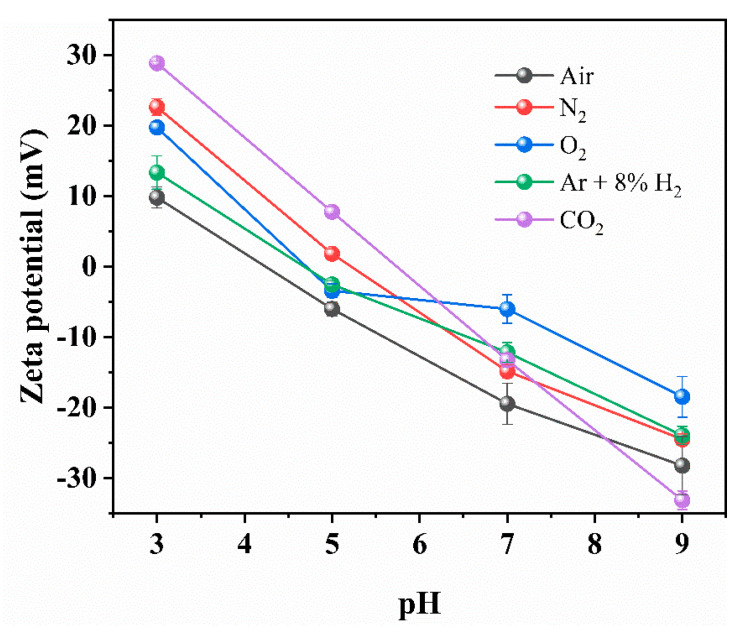
Zeta potential of N_2_, O_2_, Ar + 8%H_2_ and CO_2_ nanobubbles in deionized water as a function of pH.

**Figure 3 materials-14-01808-f003:**
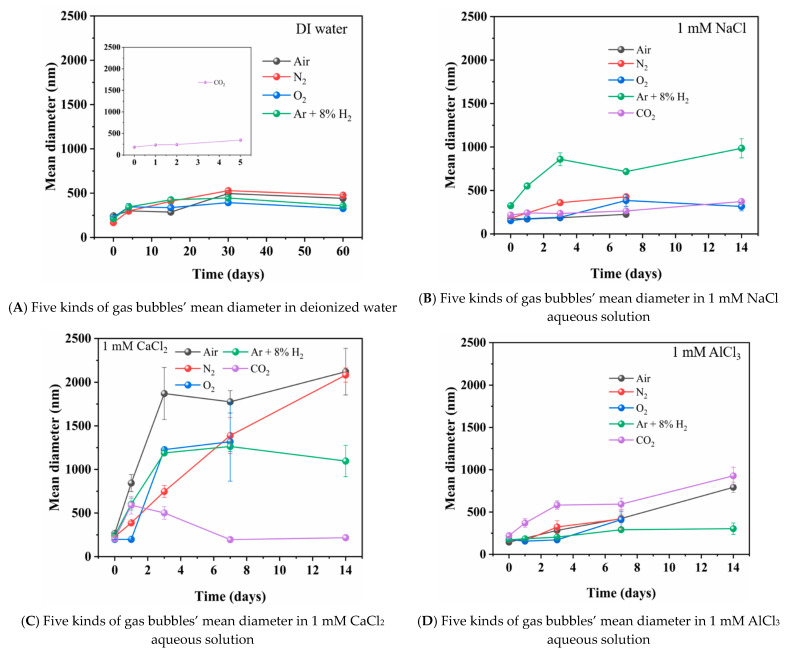
Effect of time on nanobubble mean diameter of different kind of gases (N_2_, O_2_, Ar + 8%H_2_ air and CO_2_) in (**A**) deionized water, (**B**) 1 mM NaCl aqueous solution, (**C**)1 mM CaCl_2_ aqueous solution and (**D**) 1 mM AlCl_3_ aqueous solution.

**Figure 4 materials-14-01808-f004:**
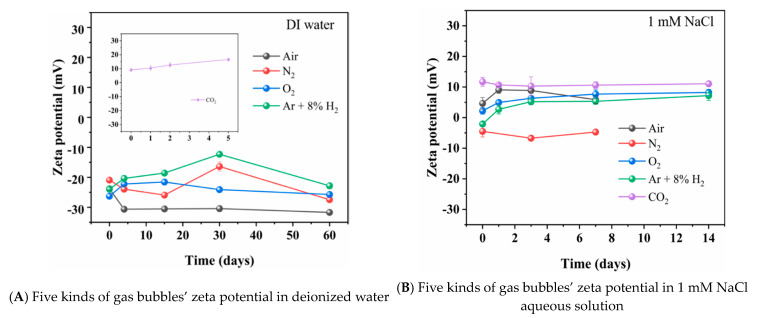
Effect of time on nanobubble zeta potential of different kind of gases (N_2_, O_2_, Ar + 8%H_2_, air and CO_2_) in (**A**) deionized water, (**B**) 1 mM NaCl aqueous solution, (**C**) 1 mM CaCl_2_ aqueous solution and (**D**) 1 mM AlCl_3_ aqueous solution.

**Figure 5 materials-14-01808-f005:**
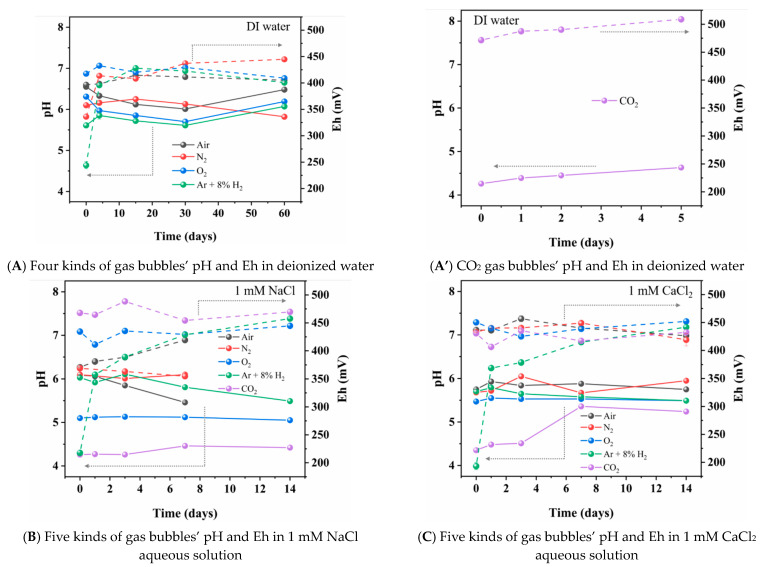
Effect of time on pH (solid line) and Eh (dashed line) of aqueous solution containing nanobubble of different kind of gases (N_2_, O_2_, Ar + 8%H_2,_ air and CO_2_) in (**A**,**A′**) deionized water, (**B**) 1 mM NaCl aqueous solution, (**C**) 1 mM CaCl_2_ aqueous solution and (**D**) 1 mM AlCl_3_ aqueous solution.

**Figure 6 materials-14-01808-f006:**
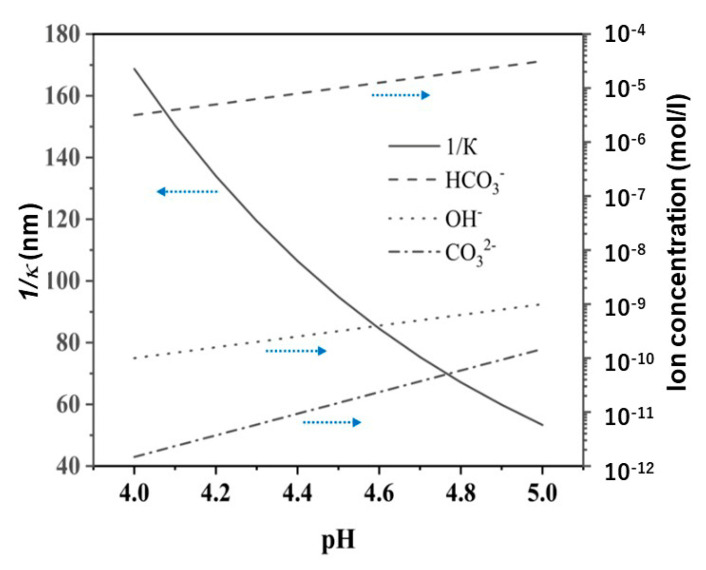
Debye length (1/*κ*) around CO_2_ bubbles (left axis) and HCO_3_^−^, OH^−^, CO_3_^2−^ ion concentration in the presence of CO_2_ nanobubbles in deionized water (right axis) as a function of pH.

**Figure 7 materials-14-01808-f007:**
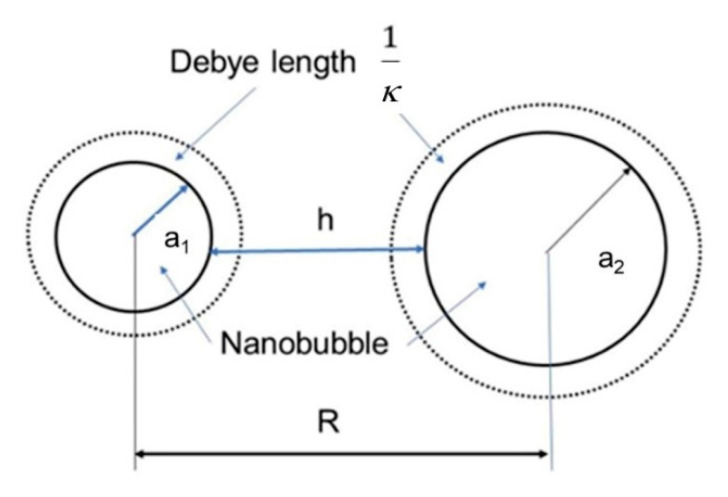
Position and geometry of two nanobubbles.

**Figure 8 materials-14-01808-f008:**
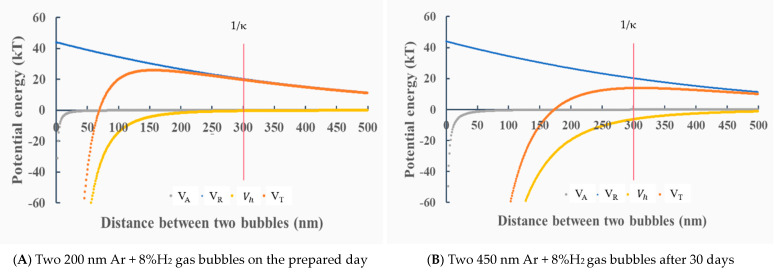
Total potential energy *V_T_* as a function of the distance between two nanobubbles at −24 mV zeta potential on the prepared day for Ar + 8%H_2_ (**A**) and −12 mV after 30 days for Ar + 8%H_2_ nanobubbles (**B**) in deionized water. (**A**) Two 200 Ar + 8%H_2_ gas bubbles on the prepared day, (**B**) two 450 nm Ar + 8%H_2_ gas bubbles after 30 days.

**Figure 9 materials-14-01808-f009:**
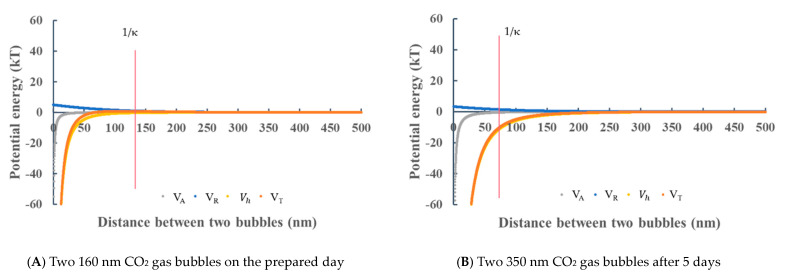
Total potential energy *V_T_* as a function of the distance between two nanobubbles at +9 mV zeta potential on the prepared day for CO_2_ (**A**) and +5 mV after 5 days for CO_2_ nanobubble (**B**) in deionized water. (**A**) Two 160 nm bubbles on the prepared day, (**B**) two 350 nm bubbles after 5 days.

**Figure 10 materials-14-01808-f010:**
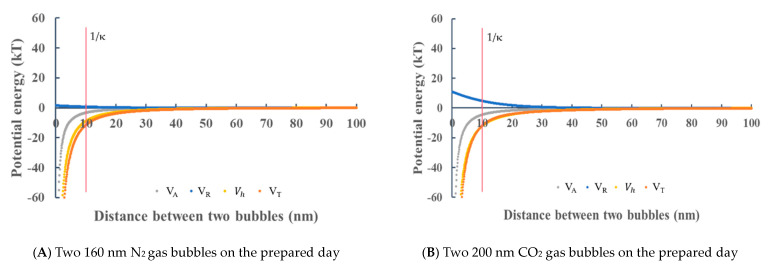
Total potential energy *V_T_* as a function of the distance between two nanobubbles at −5 mV zeta potential on the prepared day for N_2_ (**A**) and +12 mV on the prepared day for CO_2_ nanobubble (**B**) in 1 mM NaCl aqueous solution. (**A**) Two 160 nm N_2_ bubbles on the prepared day, (**B**) two 200 nm CO_2_ bubbles on the prepared day.

**Figure 11 materials-14-01808-f011:**
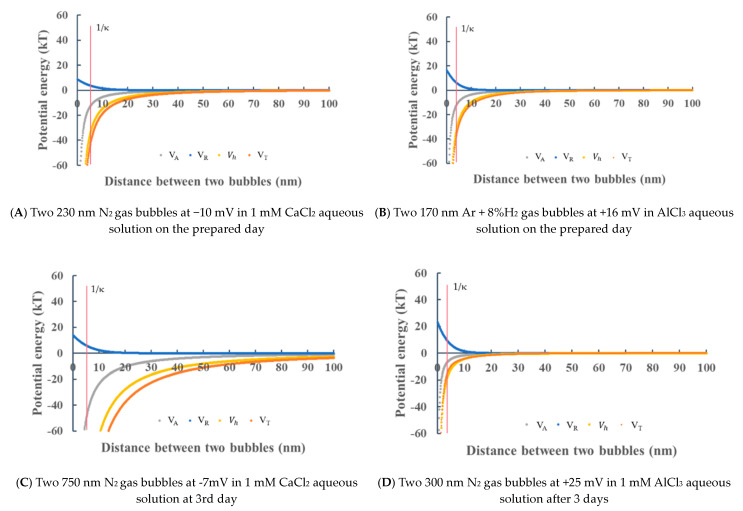
Total potential energy *V_T_* depending on the distance between two nanobubbles in the salt aqueous solution. (**A**) Two 230. nm N_2_ nanobubbles at −10mV in 1 mM CaCl_2_ aqueous solution on the prepared day, (**B**) Two 170 nm Ar + 8%H_2_ nanobubbles at +16 mV in AlCl_3_ aqueous solution on the prepared day, (**C**) Two 750 nm N_2_ gas bubbles at −7mV in 1 mM CaCl_2_ aqueous solution at 3rd day, (**D**) Two 300 nm N_2_ nanobubbles at +25 mV in 1 mM AlCl_3_ aqueous solution after 3 days.

**Figure 12 materials-14-01808-f012:**
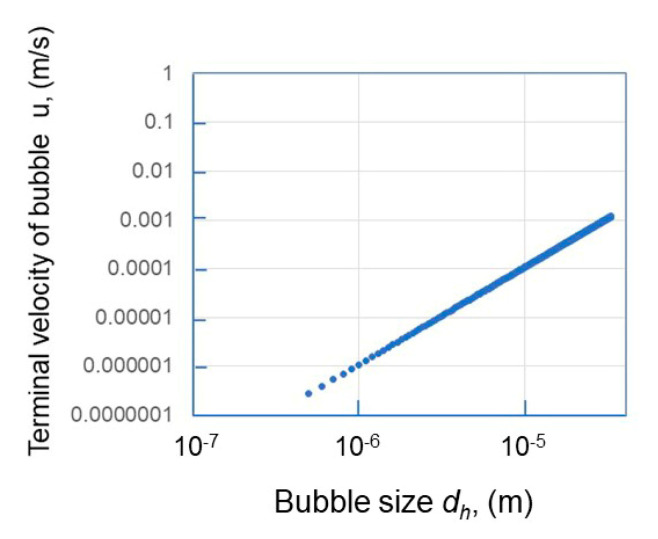
Relationship between the terminal velocity of a floating bubble and bubble size.

**Figure 13 materials-14-01808-f013:**
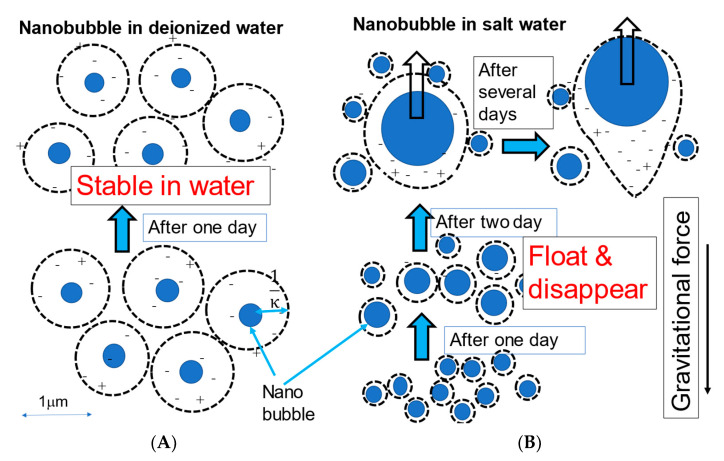
Schematic diagrams of nanobubble stability in deionized water (**A**) and nanobubble size change in salt aqueous solution (**B**) depending on time.

**Table 1 materials-14-01808-t001:** Solubility of gas, dielectric constant [37] and isoelectric point of nanobubbles determined from the results shown in Figure 1.

Gas	Solubility of Gas in 298.15 K mol gas/mol H_2_O	Dielectric Constant of Gas (Average)	Isoelectric Point (pH)
H_2_	1.455 × 10^−5^	1.0002532	
O_2_	2.748 × 10^−5^	1.0004941	4.7
8%H_2_ + Ar		1.0005247	4.6
Air		1.0005359	4.2
Ar	2.748 × 10^−5^	1.0005360	
N_2_	1.274 × 10^−5^	1.0005474	5.2
CO_2_	7.07 × 10^−4^	1.0009217	5.7

**Table 2 materials-14-01808-t002:** Stable days of gas nanobubbles produced in different solutions. The minimum days are given.

	DI Water	1 mM NaCl	1 mM CaCl_2_	1 mM AlCl_3_
Air	More than 60	7	14	14
N_2_	More than 60	7	14	7
O_2_	More than 60	14	7	7
Ar + 8%H_2_	More than 60	14	14	14
CO_2_	5	14	14	14

**Table 3 materials-14-01808-t003:** Debye length (1/*κ*, nm) of nanobubbles prepared in aqueous solutions at natural pH.

	DI Water	1 mM NaCl	1 mM CaCl_2_	1 mM AlCl_3_
N_2_	300 (pH 6)	10	5	3
O_2_	300 (pH 6)	10	5	3
Ar + 8%H_2_	300 (pH 6)	10	5	3
CO_2_	140 (pH 4.2) 70 (pH 4.7)	10	5	3

## Data Availability

The authors confirm that the data supporting the findings of this study are available within the article.

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
