# Peer review of "Long-Term Stability of Different Kinds of Gas Nanobubbles in Deionized and Salt Water"

_materials, 2021, doi:10.3390/ma14071808_

Round 1

Reviewer 1 Report

This manuscript described an experimental study on the stability of five kind s of nanobubbles ( (N2, O2, Ar+8%H2, air, and CO2) in deionized water and salt aqueous solution. The mean diameter, zeta potential of bubbles, pH, and Eh of nanobubble suspensions were measured.

The manuscript is written very clear with sufficient explanation in the introduction part. The text body is rich enough, and there is enough description regarding the experiment and results.

Meanwhile, the following comments should be addressed before publications.

  1. Is there any specific reason why the error bars in Fig(3) are so large compare to others?
  2. English language and style are fine/minor spell check required.
  3.  Equations 6, 8, 9, 11, and 12 should be typed.
  4. Figs should have a unique appearance. Y scale's colors are different for no reason. Some are black, some gray, purple, blue, and green. Some of the figs have the X-scale on top and bottom; some have only the bottom for no reason. 

Best

Author Response

Reviewer1

Thank you very much for reading our manuscript.

The manuscript was corrected as you suggested.

Meanwhile, the following comments should be addressed before publications.

  1. Is there any specific reason why the error bars in Fig(3) are so large compare to others?

As the bubbles become large, their size distribution becomes broader and thus the error bars also become larger.

  1. English language and style are fine/minor spell check required.

English was polished again throughout the manuscript.

  1.  Equations 6, 8, 9, 11, and 12 should be typed.

Yes. Equations were typed in the revised manuscript.

  1. Figs should have a unique appearance. Y scale's colors are different for no reason. Some are black, some gray, purple, blue, and green. Some of the figs have the X-scale on top and bottom; some have only the bottom for no reason. 

As suggested, in Fig.6, the y axes' colors were fixed in black colors. Fig.12 was improved and the scale showed at the bottom only.

Reviewer 2 Report

Dear Authors,

all of the comments and suggestions are listed in the attached .pdf file. I suggest that the Authors get editing help from someone with full professional proficiency in English.

Sincerely,

Reviewer

Author Response

Reviewer 2

Thank you very much for reading our manuscript.

The manuscript was corrected as you suggested.

 I found a few issues which the authors of the presented publication should refer to.

  • general remark on the equations –fractions are sometimes written in single line and sometimes in two lines. Authors probably wrote some of the equations using plain text and the rest of them using equation editor. All of the equations should be written in the same form.

As suggested, all the equations were typed and improved in the same form in the revised manuscript.

  • general remark on the Tables and Figures –some of the Tables or Figure captions have been split into two pages

All the figures and tables were fixed in the same page in the revised manuscript.

  • 1, lines 19-21: Authors wrote that “The nanobubbles except CO2 in deionized water showed the long-term stability for 60 days, while they were not stable in 1 mM (milli mol/L) salt aqueous solution”, but on p.2 it is said, that “The low concentration 1 mM (milli mol/L) of NaCl can stabilize O2 micro-and nano-bubbles(...)”.

The abstract in p.1 summarized our own results while the introduction in p.2 described literature results. In order to differentiate them clearly, the statement in p.2 was modified as follows by mentioning one-week stability duration from reference [7].

  The low concentration 1 mM (milli mol/L) of NaCl could stabilize O2 micro- and nano-bubbles for at least one week [7] because there was a shielding effect in low concentration of NaCl.

4) p. 3, line 82: “Deionized water with a conductivity of 18.2 MΩcm”–

I believe this is the resistivity, not conductivity of DI water.

It was changed according to your suggestion.

p.3 line 84,  conductivity → resistivity

  • 3, lines 103-105: “The machine worked for 15 minutes to generate high number density nanobubble water of more than 108 particles/mL (obtained from NanoSight, NS300,Malvern)”–why the number density was not determined for all the nanobubble suspensions along with the mean diameter? The change in the number of nanobubbles per volume unit and its mean diameter could support the assumptions of Ostwald ripening or floating and collapse of the nanobubbles at the air-water interface

Our laboratory has only DLS, nano-Book Omni America. There is no equipment of NanoSight in our University and we ordered one measurement by NanoSight outside to know the number density of nanobubble. Therefore, the word “outsourcing” was added as follows in the revised manuscript.

.p.3  line 107,

・・・to generate high number density of nanobubble in water of more than 108 particles /mL (obtained from NanoSight, NS300, Malvern, outsourcing)

.6) p. 4, line 111: “(...)temperature of about 40°C.”–in the manuscript.pdf file the degree Celsius symbol is displayed incorrectly. Similar issues can be found later in the text for other symbols ,e.g. p. 5, Equations (3) and (5); p. 6, Equation (7). Authors probably used different font for some symbols than in the plain text.

In order to address the issues you kindly pointed out, all the temperature descriptions were converted from Celsius (oC) to Kelvin (K) as used in Table 1.

oC  → K

7) p. 5, lines 138-139:“(...)Smolukowski model”–it should be “Smoluchowski model”.

p.5 line 138

It was changed as suggested in the revised manuscript.

Smolukowski model → Smoluchowski model

8) p. 13, line 308:“Table 1.”–it should be “Table 2.”.

The Table number was corrected as suggested.

Table 1 →Table 2

9) p. 14,Eq.(11)line 353and Eq. (12)line 359–authors should use subscript in “a1”and “a2”symbols.

As suggested, the all numbers in a1 and a2 changed to subscript in a1 and a2

a1, a2→ a1, a2

10) p. 15, line 362:“Table 2.”–it should be “Table 3.”.

p.14 line 342,345,346

The Table number and sentences were corrected, as you kindly pointed out.

Table 2 →Table 3

11) p. 20, line 474: “(...)the bubble movement distance is Dx”–it should be the root mean square of bubble displacement√?2Ì…Ì…Ì….

As suggested, now the following changes were applied in p.20-21, line 499-501 of the revised manuscript.

Bubble displacement is shown in

In equation (14)    = t = h

In equation (15)    =

12) p. 20, lines 486-488: “According to our calculation, a bubble larger than about 0.1 μm (100 nm) can gradually float and disappear at the air-water interface”–the bubble diameter of 100 nm was assumed from the intersection of two lines in Fig. 12.

Why not calculate the exact value comparing Eqs. (15) and (16)? But, when I calculated the hydrodynamic diameter for the following parameters:?=0.001??∙??=1.38∙10−23?/?

?=293??=998??/?3?=1? calculated value of hydrodynamic diameter was equal to ?â„Ž=1.24??. I also recreated calculations which results are shown in the Fig. 12 in the Manuscript (see Fig. A below): Figure A. Distance travelled by the bubble in 1 second due to the floating and free diffusion Results presented in the Fig. A. disagree with the assumptions made by the Authors(p. 20, lines 483-490).Moreover, the “bubble diffusion distance Δx”calculated using Eq. (15) is the mean length of the path travelled by the bubble. Authors should consider that the diffusion pathway of molecules is (pseudo) random. Mean free path of water molecules at STP is about 0.25 nm(2.5‧10-7mm), which is several orders of magnitude less than the calculated “bubble diffusion distance Δx”.Particle (bubble)undergoing diffusion change movement direction many times in one second. On the other hand, displacement of the bubble due to buoyancy force is always pointing in the same direction (upwards). Authors should distinguish the root mean square displacement of the particle(calculated using Eq. (15)), which is the mean length of the path travelled by the particle, and mean displacement of the particles undergoing free diffusion, which is equal to zero (see: Berg, H. C. (1993). Random walks in biology, Chapter 1,Diffusion: Microscopic theory. Princeton University Press).Brownian motion can influence the residence time distribution of the suspended bubbles, but no matter the bubble diameter is, it always tends to travel upwards due to buoyancy.

Thank you very much for your suggestions.

Fig. 12 was changed to the Relationship between the terminal velocity of a floating bubble and bubble size in the revised manuscript.

The sentences were corrected as follows,

p.21 line 508-513

Bubbles experiencing diffusion change their movement direction frequently in short time. On the other hand, the bubble displacement due to the buoyancy force is always pointing upwards against the gravitational force. The terminal velocity depends on the bubble diameter as shown in Fig.12, based on our calculation using equation (16). A bubble smaller than 1 mm can experience very low terminal velocity (< 1 x 10-6 m/s) that prevents bubbles to float against the buoyancy force. Our results partially agree with previous literature.

Fig.12. Relationship between the terminal velocity of a floating bubble and bubble size.

Reviewer 3 Report

The authors did a systematic study on the stability of five different gas nanobubbles in the aqueous solutions. In the manuscript, the stability of nanobubbles was predicted using the DLVO theory, in which the total interaction energy between the two bubbles consisted of the van der Waals and electrical double layer potentials. Although the predictions were in agreement with the experimental results, the reviewer still does not recommend for publication due to the following issues:

Major:

  1. It is generally accepted that air bubble are hydrophobic entities in water. Many researchers reported that hydrophobic interactions existed in the foam film between two air bubbles. Therefore, the DLVO theory may not be directly applied for predicting the stability of nanobubbles. Here are some related references showing the hydrophobic interactions in the bubble-bubble interactions,

Tchaliovska et al., Journal of Colloid and Interface Science, 1994, 168, pp.190-197

Angarska et al., Langmuir, 2004, 20, pp.1799-1806

Wang and Yoon, Colloids and Surfaces A: Physicochem. Eng. Aspects, 2005, 263, pp.267-274

Minor:

  1. The quality of the figures needs improvement in the manuscript. In Figs. 3, 4, 5, 8, 9, 10, and 11, each sub-figure should have a title so that the readers can understand them better. Also, too many lines are plotted in Fig. 5, which makes it hard to read. In Fig. 11, it is not necessary to mention the bubble size because van der Waals and EDL interaction energies will not change with particle size when the two bubbles are of the same size.
  2. The authors did not describe how to measure the Eh values in the manuscript.
  3. The English needs improvement in this manuscript.

Author Response

Reviewer 3

Thank you very much for reading our manuscript.

The manuscript was corrected as you suggested.

Comments and Suggestions for Authors

Major:

  1. It is generally accepted that air bubble are hydrophobic entities in water. Many researchers reported that hydrophobic interactions existed in the foam film between two air bubbles. Therefore, the DLVO theory may not be directly applied for predicting the stability of nanobubbles. Here are some related references showing the hydrophobic interactions in the bubble-bubble interactions,

As suggested, the extended DLVO theory was used by adding the hydrophobic interaction in the revised manuscript, instead of the DLVO theory.

p.14 line 350-355, the following sentences were added.

Tchaliovska et al. investigated the thickness of thin flat foam films formed in aqueous dodecyl ammonium chloride solution [44]. Angarska et al. showed that the value of the critical thickness of foam film was sensitive to the magnitude of the attractive surface forces acting on the film using sodium dodecyl sulfate and the total disjoining pressure operative on the films was expressed as a sum of the van der Waals and hydrophobic contributions [45].

[44] S. Tchaliovska, E. Manev, B. Radoev, J. C. Eriksson, P.M. Claesson, Interactions in equiribrium free films of aqueous dodecyl ammonium chloride solutions, Journal of colloid and Interface Science 168 (1994) 190-1944.

[45]  J. K. Angarska, B.S. Dimitrova, K.D. Danov, P. A. Kralchevsky, K. P. Ananthapadmanabhan, A. Lips, Detection of the hydrophobic surface force in foam films by measurements of the critical thickness of the film rupture, Langmuir, 20 (2004) 1799-1806.

p.14 line 360-363

The equations of (10) and (11) were modified in the following manner according to the extended DLVO theory. Also, all the potential calculation results shown in Figs. 8,9,10,11 were updated in the revised manuscript.

The total potential energy VT between two nanobubbles can be given by the potential energy due to the van der Waals interaction (VA), hydrophobic interaction (Vh) and the electrostatic interaction (VR) as follows,

 line 360    equation      (10)

 line363     equation      (11)

p.15 line 366-373, The following sentences are added.

The K232 is a hydrophobic constant for air 2 in water 3. Yoon and Aksoy calculated the K232 using the extended DLVO theory [49] while Wang and Yoon measured the K232 as a function of SDS concentration at different kinds of NaCl concentration and showed K232 was estimated 10-17 J in the absence of salt and 10-19 J in 1mM NaCl aqueous solution [50]. In our calculation, 10-19 J of K232 was used in 1 mM CaCl2 and AlCl3 aqueous solutions and 10-18 J of K232 was used in deionized water containing CO2 nanobubbles by considering the reference values [50].

[49] R. H. Yoon, B. S. Aksoy, Hydrophobic forces in thin water films stabilized by dodecyl ammonium chloride, Journal of Colloid and Interface Science, 211 (1999)1-10.

[50] L. Wang, R.H. Yoon, Hydrophobic forces in the foam films stabilized by sodium dodecyl sulfate: Effect of electrolyte, Langmuir, 20 (2004), 11457-11464.

Minor:

  1. The quality of the figures needs improvement in the manuscript. In Figs. 3, 4, 5, 8, 9, 10, and 11, each sub-figure should have a title so that the readers can understand them better. Also, too many lines are plotted in Fig. 5, which makes it hard to read. In Fig. 11, it is not necessary to mention the bubble size because van der Waals and EDL interaction energies will not change with particle size when the two bubbles are of the same size.

Thank you for your suggestions.

In Figs. 3, 4, 5, 8, 9, 10, and 11,

each sub-figure and the sub-title were added and the quality of the figures was tried to improve further for a reader to understand them better.

In Fig.5

the arrows were shown in the dashed line and solid line to point out which dataset corresponds to which value.

In Fig.11

the calculation results for same size bubbles was indicated. Our intention is to evaluate the stability of different nanobubbles in the presence of different salts as a function of time after their preparation. Our calculations were based on our experimental results of size and zeta potential measurements of nanobubbles. The calculation results shown in Fig. 11 are not simple comparison of bubble size effect on the total potential energy, but compare nanobubble stability as a function of different origin and conditions.

2. The authors did not describe how to measure the Eh values in the manuscript.

p.5 line 149-150, the following information was added in the revised manuscript.

Eh meter (by oxidation-reduction potential meter, Mettler Toledo, Inlab Redox )

3. The English needs improvement in this manuscript.

English writing was polished again throughout the manuscript.

Round 2

Reviewer 2 Report

Thank You for addressing all of my comments. In the revised version of the manuscript "Smoluchowski model" is still misspelled (line 136). 

I found no major issues and in my opinion the article may be published.

Reviewer 3 Report

The authors have addressed the major and minor issues. I suggest publication in Materials.